# Serum albumin and gamma gap levels, and combined effect for risk of mortality in a Japanese population from the J-MICC study

Kenichi Shibuya[1,2], Rie Ibusuki[3], Daisaku Nishimoto[4], Shiroh Tanoue[5], Chihaya Koriyama[5], Shuhei Niiyama[1], Yasuyuki Kakihana[1], Toshiro Takezaki[6]*, Megumi Hara[7], Yuichiro Nishida[7], Sadao Suzuki[8], Takeshi Nishiyama[8], Mako Nagayoshi[9], Takashi Tamura[9], Yudai Tamada[9], Rieko Okada[9], Teruhide Koyama[10], Satomi Tomida[10], Kiyonori Kuriki[11], Jun Otonari[12], Hiroaki Ikezaki[13], Asahi Hishida[14], Masashi Ishizu[15], Sakurako Katsuura-Kamano[15], Kenji Wakai[9], Keitaro Matsuo[16,17], for the J-MICC Study group†

1 Department of Emergency and Intensive Care Medicine, Kagoshima University Graduate School of Medical and Dental Sciences, Kagoshima, Japan, 2 Kagoshima Prefectural Tokunoshima Public Health Center, Ohshima, Japan, 3 Department of Community-Based Medicine, Kagoshima University Graduate School of Medical and Dental Sciences, Kagoshima, Japan, 4 School of Health Sciences, Faculty of Medicine, Kagoshima University, Kagoshima, Japan, 5 Department of Epidemiology and Preventive Medicine, Kagoshima University Graduate School of Medical and Dental Sciences, Kagoshima, Japan, 6 Community Medicine Support Center, Kagoshima University Hospital, Kagoshima, Japan, 7 Department of Preventive Medicine, Faculty of Medicine, Saga University, Saga, Japan, 8 Department of Public Health, Nagoya City University Graduate School of Medical Sciences, Nagoya, Japan, 9 Department of Preventive Medicine, Nagoya University Graduate School of Medicine, Nagoya, Japan, 10 Department of Epidemiology for Community Health and Medicine, Kyoto Prefectural University of Medicine, Kyoto, Japan, 11 Laboratory of Public Health, Division of Nutritional Sciences, School of Food and Nutritional Sciences, University of Shizuoka, Shizuoka, Japan, 12 Department of Psychosomatic Medicine Graduate School of Medical Sciences, Kyushu University, Fukuoka, Japan, 13 Department of General Internal Medicine, Kyushu University Hospital, Fukuoka, Japan, 14 Department of Public Health, Aichi Medical University, Nagakute, Japan, 15 Department of Preventive Medicine, Tokushima University Graduate School of Biomedical Sciences, Tokushima, Japan, 16 Division of Cancer Epidemiology and Prevention, Aichi Cancer Center, Nagoya, Japan, 17 Department of Cancer Epidemiology, Nagoya University Graduate School of Medicine, Nagoya, Japan

† Complete membership of the J-MICC Study Group is listed in the Acknowledgments.
* takezaki@m.kufm.kagoshima-u.ac.jp

## Abstract

Although the clinical importance of serum albumin and gamma gap levels is well established, it is unclear how these levels are associated with health risks in the general population. This cohort study aimed to clarify the association between serum albumin and gamma gap levels, and their combined effect, and mortality risk in a Japanese population. The participants totaled 35,746 (17,160 men and 18,586 women) aged 35–69 years from the Japan Multi-Institutional Collaborative Cohort (J-MICC) Study. The mean follow-up period was 11.8 years, with 1,529 deaths and 1,907 censoring. The Cox proportional hazards model was used to estimate hazard ratios (HRs) and 95% confidence intervals after adjusting for related factors. Increased HRs of low albumin and high gamma gap levels were respectively observed for deaths from all-causes, cancer, cardiovascular diseases, respiratory

**Data availability statement:** The data used in this study cannot be made publicly available due to ethical restrictions. Our J-MICC study had the informed consents from the participants on data availability, which all data should be used within the J-MICC study group, and ethics review is required when the data is provided to another study. The data is available upon reasonable request from the J-MICC Central Office (wakai.kenji.y2@f.mail.nagoya-u.ac.jp) in the Department of Preventive Medicine, Nagoya University Graduate School of Medicine, Nagoya, Japan.

**Funding:** This study was supported by Grants-in-Aid for Scientific Research for Priority Areas of Cancer [grant number 17015018], Innovative Areas [grant number 221S0001], and the Japan Society for the Promotion of Science (JSPS) KAKENHI [grant numbers. 16H06277 and 22H04923; CoBiA] from the Japanese Ministry of Education, Culture, Sports, Science, and Technology. The funders had no role in study design, data collection and analysis, decision to publish, or preparation of the manuscript.

**Competing interests:** No authors have competing interests.

system diseases without pneumonia, and other-causes; and the HR was the highest on respiratory system diseases without pneumonia (HR = 7.31, 4.15–12.9). Low albumin and low gamma gap levels were strongly associated for pneumonia death (HR = 12.4, 3.98–38.5). The interaction between albumin and gamma gap levels was significant for deaths from all-causes, pneumonia and other-causes. The dose relationship for each association was dose-dependent in albumin and threshold-type in gamma gap, except for other-causes. This study suggests that albumin and gamma gap levels are independent indicators of an increased risk of mortality in a Japanese population. Combined effect was apparent for mortality from all-causes, pneumonia, and other-causes.

## Introduction

Serum total protein is the sum of hundreds of proteins. Albumin makes up more than half of them, and the remainder is gamma gap. Their clinical significance at low and high levels has been established [1]. Furthermore, it is widely used in health checkup items for the general population although its significance from the standpoint of preventive medicine has not been fully elucidated [1].

Serum total protein reflects complex pathways such as ingestion, absorption, synthesis, catabolism, and leakage [1]. Low serum total protein levels are observed in diseases including nephrotic syndrome and severe liver disorders such as liver cirrhosis, and malnutrition. High levels are evident in sarcoidosis, multiple myeloma, and dehydration. As the serum total protein level is the sum of serum albumin and gamma gap levels, it is inferior in sensitivity and specificity compared to albumin and gamma gap alone as an indicator of disease or pathophysiology [1].

Albumin, which constitutes most of the serum total protein, is synthesized in the liver after dietary proteins are broken down into amino acids in the small intestine and absorbed by the epithelial mucosa of the small intestine. After that, it is decomposed in locations such as muscles, skin, liver, kidneys [2]. Therefore, in addition to insufficient nutritional intake, serum albumin levels decline due to impaired absorption of amino acids because of a breakdown in the gastrointestinal mucosal epithelium, decreased synthesis capacity in the liver from cirrhosis, protein leakage from the gastrointestinal tract and kidneys due to protein-leaking gastroenteropathy and nephrotic syndrome, and hyper-catabolism from severe infections and chronic inflammatory diseases. Among them, production disorder has a great influence on a serum albumin decrease, which depends on the ability of the liver to synthesize [3].

Albumin is a negatively charged protein with a high concentration and relatively large molecular weight, accounting for approximately 75–80% of plasma colloid osmotic pressure [2]. The negative charge of the albumin molecule also attracts cations and water, contributing to water retention within capillaries and playing a crucial role in stabilizing blood volume [2]. Furthermore, albumin molecules possess various ligand-binding sites, enabling them to bind and transport substances such as fatty acids, hormones, bilirubin, and calcium [2]. Approximately 40% of the calcium in

whole blood exists in a bound state with albumin, which buffers and stabilizes blood calcium levels [1]. Highly hydrophobic molecules also become soluble in blood by binding to albumin, allowing them to be transported to target tissues [2]. Furthermore, albumin binds to drugs and toxins, suppressing the concentration of free active forms and mitigating rapid onset of action or toxicity [2]. Physiologically, low albumin levels impair the maintenance of adequate plasma colloidal osmolarity. This makes it easier for fluid to leak from blood vessels into tissues, potentially causing edema, pleural effusion, and ascites. Furthermore, as albumin transports various substances, its reduction serves as an indicator of conditions such as malnutrition, liver dysfunction, and chronic inflammation. Clinically, hypoalbuminemia is a poor prognostic indicator, with numerous studies showing a strong association with increased mortality in hospitalized patients and those with chronic diseases [4,5].

In contrast, elevated serum albumin levels are noted during dehydration. The main role of albumin in the body is to maintain plasma oncotic pressure; bind to nutrients such as fatty acids, endogenous substances such as bilirubin, and exogenous substances such as drugs; and contribute to their transport, solubilization, and stabilization [6].

Gamma gap which is the difference between total serum protein and serum albumin is divided into four fractions of α1, α2, β, γ globulins [1]. The α1 and α2 fractions contain most of the acute phase reaction proteins, and these fractions are increased in inflammatory diseases. The main fraction of β fraction is transferrin, and this fraction is elevated in iron deficiency anemia, and during pregnancy. Gamma-globulin is produced from plasma cells that arise from B cells, which are produced in the bone marrow. The gamma fractions contain five immunoglobulins (Ig), IgG, IgA, IgM, IgD, and IgE, which play important roles in the immune system as antibodies. A gamma gap is used as a clinical screening index for conditions such as latent inflammation, cancer, autoimmune diseases [7–10]. Physiologically, a high gamma gap suggests increased immunoglobulin levels. This indicates the possibility of ongoing chronic inflammation within the body or abnormal plasma cell proliferation. Clinically, it can serve as a differential diagnostic indicator for conditions such as multiple myeloma, macroglobulinemia, and primary amyloidosis. It is also used as a screening indicator for plasma cell disorders, and when a marked elevated gamma gap is observed, further investigation such as protein electrophoresis is recommended.

Conversely, a decrease in blood globulin levels is observed during congenital hypogammaglobulinemia, multiple myeloma, malignant lymphoma, and drug-induced reactions to immunosuppressants and steroids [1].

During medical examinations for the general population, healthy examinees can have slightly low or high serum total protein and albumin levels, and albumin/globulin (A/G) ratios; however, since they are part of the general population, it is rare for them to have an underlying disease that clinically exhibits these abnormal values. To the best of our knowledge, there is less evidence for how low or high levels of serum total protein, albumin and gamma gap are associated with health risks in the general population, other than two study on low albumin levels [11,12] and one study on gamma gap [13]. Furthermore, no study investigated their combination effect. As albumin plays an important role in binding to various substances, their transport, solubilization, and stabilization [6], and gamma gap is related to latent inflammation, cancer, autoimmune diseases [7–10], their interction may exist.

To clarify the association between the levels of serum albumin and gamma gap, and their combined effect, and mortality risk, this study conducted a cohort study among a Japanese population who participated in the Japan Multi-Institutional Collaborative Cohort (J-MICC) study.

## Methods

### Participants

The participants were recruited from the J-MICC study. The J-MICC study was launched in 2005 and participant recruitment started on October 12, 2005, except one cohort field of Kyushu University which recruitment started from October 1, 2004 and was included in the J-MICC study group in 2005. Its details have been described elsewhere [14–16]. The candidates are those aged 35–69 years at enrollment from 13 cohorts in Japan (n = 92,514), whose resident records are at the local government offices of the target areas. They completed questionnaires that assessed their lifestyle behaviors,

and provided physical data and blood samples at baseline. The present study used the dataset fixed on October 16, 2023, version 20231016, which recruitment period of participants was from October 1, 2004 to March 7, 2014.

The present study comprised a total 35,853 individuals from 10 cohorts, whose examination data of both total protein and albumin were available. Then, 107 participants were excluded according to the following criteria: recoding mistake of total protein and/or albumin (n = 1); and dead cases within two years from the baseline (n = 106). Finally, 35,746 eligible participants (17,160 men and 18,586 women) were included in this study (Fig 1).

This study was conducted in accordance with the Declaration of Helsinki, and the study protocol was approved by the Ethics Review Committee for Human Genome/Gene Analysis Research at Kagoshima University Graduate School of Medical and Dental Sciences (No. 16, 450 and 487), the Expert Committee on Ethics Review of Human Genome and Gene Analysis at Kyushu University Graduate School of Medicine (No. 131), Observation Research Ethics Review Committee (No. 590−04), and the ethics review boards of all institutions and universities participating in the J-MICC study. All the participants provided written informed consent.

## Lifestyle factors

A structured self-administered questionnaire was used to obtain lifestyle information, and the contents and procedures were standardized in the J-MICC study regions. The questionnaire comprised items regarding sociodemographic

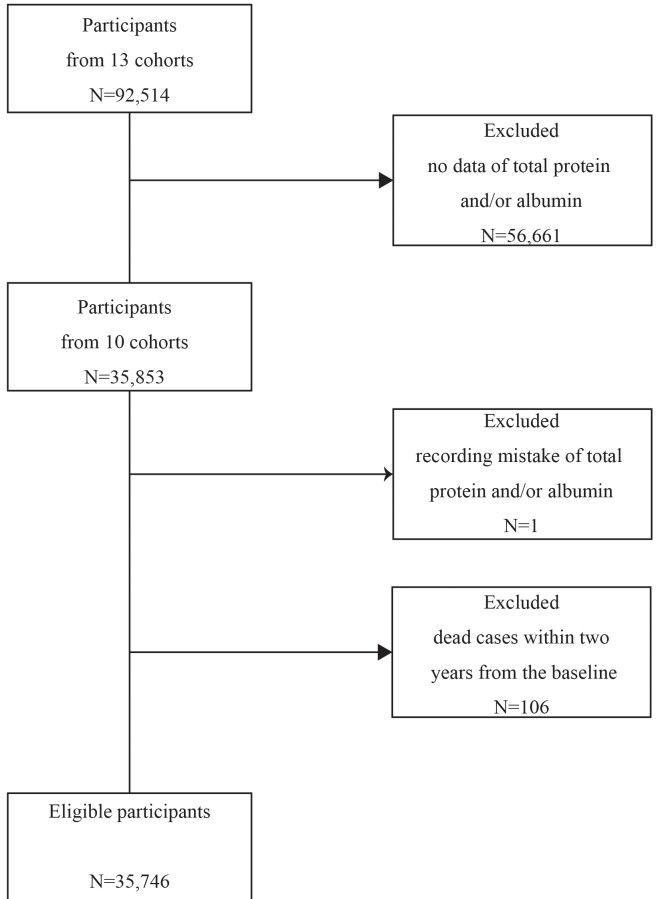

**Fig 1. Flow diagram for selecting eligible participants from 13 cohorts.**

characteristics, smoking, drinking and dietary habits, daily activities, habitual exercise, sleeping situation, stress status, personal and family medical history, intake of prescription medicines and supplements, and reproductive history.

Information on dietary habits was obtained for average intake during the past year using the food frequency questionnaire (FFQ), which includes three staples, forty-one food items, and nine beverages [17–19]. The average intake of staple foods was also assessed in addition to their frequency. Total energy and protein intakes were estimated using the FFQ. Although the FFQ did not capture the actual total energy intake, previous validity studies have shown that the estimated energy and protein intakes using the present FFQ was available for comparison [17–19].

## Clinical characteristics

Clinical data included systolic blood pressure (SBP), diastolic blood pressure (DBP), triglycerides (TG), total cholesterol, high-density lipoprotein cholesterol (HDL-C), fasting blood glucose (FBG), serum total protein, and serum albumin. Examination of serum total protein levels was carried out using the Biuret method. Serum albumin levels were examined using bromocresol green (BCG) test or modified bromocresol purple (BCP) test. Biochemical tests were performed for each study region. Low-density lipoprotein cholesterol (LDL-C) levels were calculated according to the Friedewald formula using a TG level <400 mg/dL [20].

## Follow-up

Follow-up information was obtained from the date of the baseline survey to the final date of the follow-up. Participants who moved out of the study regions or were denied authorization by a government official were censored. The duration of follow-up was calculated as the time from the date of a participant's baseline survey to their death, the censored date, December 31, 2021, or December 31, 2020, according to the follow-up procedures for each study region, whichever came first. During an average follow-up of 11.8 years, 1,529 people died, and 1,907 people were censored. The number of deaths from cancer (International Classification of Diseases, Tenth Revision [ICD 10]: C00-D48), cardiovascular diseases (CVD) (ICD 10: I10-I99), respiratory system diseases (RSD) without pneumonia (ICD 10: J00–J06, J30-J99), pneumonia (ICD 10: J09–J22) and other causes were 769, 263, 104, 33 and 360, respectively.

## Statistical analyses

Age was divided into three groups (35–49, 50–59, and 60–69 years) according to similar number of participants. Smoking and drinking habits were categorized as current or former vs. never. Total alcohol consumption was estimated as the accumulated intake amount using the frequency and amount of each beverage type among current drinkers. Body mass index (BMI) was categorized as < 18.5 kg/m$^2$, 18.5–24.9 kg/m$^2$, and ≥ 25 kg/m$^2$. Metabolic equivalents (METs) for habitual exercise and daily activity were calculated, accounting for intensity, frequency, and duration, as reported in the questionnaire [21,22]. The estimation of METs h/day was then categorized into three groups according to a similar number of participants. Energy-adjusted intakes of protein was estimated using the residual method [23]. The energy-adjusted intake of protein was also categorized into three groups as < 49.8, 49.8–55.2, and > 55.2 g/day according to a similar number of participants.

Hypertension was defined as SBP ≥ 140 mmHg, DBP ≥ 90 mmHg, or the intake of antihypertensive medications. Dyslipidemia was defined as TG level ≥ 150 mg/dL, LDL-C level ≥ 140 mg/dL, or HDL-C level < 40 mg/dL or the use of lipid-lowering agents. Glucose intolerance was defined as an FBG level ≥ 110 mg/dL or the use of antidiabetic medication.

Serum total protein and albumin levels comprised three groups, using cutoff values of the Japanese Committee for Clinical Laboratory Standards: < 6.6, 6.6–8.1 and >8.1 g/dL for serum total protein; and <4.1, 4.1–5.1 and >5.1 g/dL for albumin [24]. Gamma gap level was calculated as "(total protein) – (albumin)" and its cutoff values were also estimated using each cutoff value of total protein and albumin as <2.5, 2.5–3.0 and >3.0 g/dL.

Person-years were calculated from the date of baseline to the date of death, date censored, or last date of follow-up, whichever occurred first. Cases of death within two years from the baseline survey were excluded from the analysis to minimize potential reverse causality effects. HRs and their 95% confidence intervals (CIs) for death according to total protein, albumin and gamma gap levels were estimated using the Cox proportional hazard model after adjusting for confounding factors in three models: model 1: adjusted for age, sex and BMI; model 2: adjusted for age, sex, BMI, smoking, drinking, daily activity, habitual exercise, education, intake of meat, fish, tofu, green vegetables, fruit and coffee, hypertension, dyslipidemia, glucose intolerance, and history of peptic ulcer, gastritis, colon polyp, liver cirrhosis, fatty liver, chronic bronchitis, asthma and cancer; and model 3: adjusted for the same variables of model 2 adding proteinuria and energy-adjusted protein intake instead of meat, fish, and tofu intakes. The variables used for adjustment were categorized into two or three groups according to a similar number of participants. HRs for death were also estimated according to the combination of low/medium/high albumin and gamma gap levels. $P$-value for interaction between each two group of albumin and gamma gap was also calculated using each medium-level group as reference. Furthermore, subgroup analysis was done among those without having history of diseases to observe the association impact after reduced potential reverse causality effects by history of diseases that cannot be controlled by the adjustment.

As some of the factors related to higher levels and those related to lower levels in albumin and gamma gap may be potentially different, restricted cubic splines (RCSs) were used to access the non-linear association between the HRs and these levels. A spline curve combined cubic polynomials and linear terms was expressed using five knots and reference. Each HR and 95% CI were estimated after adjusted variables in the model 3, and these variables were centered before the adjustment.

The relationship between the baseline characteristics and those with low and high levels of serum albumin and gamma gap at baseline was tested. Odds ratios (ORs) and 95% CIs were obtained by multivariate logistic regression analysis using variables of model 2 and 3 as covariates. The relationship among serum total protein, albumin, and gamma gap levels were also tested using correlation coefficients.

Differences were considered significant for p-values < 0.05. All statistical analyses were performed using Stata software (version 15; Stata Corp., College Station, TX, USA).

## Results

The percentage of the participants at a higher age group was higher than that at a lower age group in both men and women (Table 1). Obesity with a BMI ≥ 25.0, current smokers, and current drinkers were more common in men. The percentages of participants with low and high serum total protein levels were 7.6% and 2.8% in men and 5.1% and 3.9% in women, respectively. Those with low serum albumin and gamma gap levels were 11.3% and 17.8% in men and 12.6% and 10.3% in women, respectively. Those with high levels of serum albumin and gamma gap were 0.4% and 19.9% in men and 0.4% and 28.0% in women, respectively. Proteinuria was observed in 8.3% of men and 4.0% of women. The mean levels of serum total protein, albumin and gamma gap in men and women were 7.22 and 7.30; 4.45 and 4.43; and 2.77 and 2.87, respectively. The correlation coefficients between serum total protein levels, and albumin and gamma gap levels were 0.523 (p < 0.001) and 0.777 (p < 0.001) in men, and 0.536 (p < 0.001) and 0.794 (p < 0.001) in women, respectively (data not shown in Table). The correlation coefficients between albumin and gamma gap levels were −0.130 (p < 0.001) in men and −0.088 (p < 0.001) in women.

In the cross-sectional analysis at baseline, increased ORs for low albumin levels were observed in the older age, men, high BMI, current smoking, frequent tofu intake, history of peptic ulcer, colon and liver cirrhosis, and proteinuria groups; decreased ORs were observed in low BMI, coffee intake, hypertension, dyslipidemia, glucose intolerance, and history of fatty liver after adjusting for related factors (Table 2). The impact of increased OR for low albumin levels was highest on liver cirrhosis history (3.33, 2.10–5.28). Increased ORs for high albumin levels were observed in current drinking, hypertension, dyslipidemia, history of fatty liver, whereas decreased ORs were observed for men. The

**Table 1. Characteristics of the study participants by sex at the baseline.**

| | Men | | Women | |
|---|---|---|---|---|
| | N | % | N | % |
| Age (years) | | | | |
| 35-49 | 4,525 | 26.4 | 5,645 | 30.4 |
| 50-59 | 5,631 | 32.8 | 6,278 | 33.8 |
| 60-69 | 7,004 | 40.8 | 6,663 | 35.8 |
| Total | 17,160 | 100 | 18,586 | 100 |
| BMI | | | | |
| <18.5 kg/m² | 476 | 2.8 | 1,591 | 8.6 |
| 18.5–24.9 kg/m² | 11,630 | 67.8 | 13,495 | 72.6 |
| ≥25.0 kg/m² | 5,052 | 29.4 | 3,494 | 18.8 |
| Smoking (current) | 4,902 | 28.6 | 1,259 | 6.8 |
| Drinking (current & ≥20 g alcohol/day) | 6,625 | 38.6 | 969 | 5.2 |
| Energy-adjusted protein intake | | | | |
| <49.8 g/day | 5,823 | 33.9 | 6,092 | 32.8 |
| 49.8–55.2 g/day | 5,447 | 31.7 | 6,468 | 34.8 |
| >55.2 g/day | 5,890 | 34.3 | 6,026 | 32.4 |
| Serum total protein level | | | | |
| <6.6 g/dL | 1,310 | 7.6 | 949 | 5.1 |
| 6.6–8.1 g/dL | 15,367 | 89.6 | 16,919 | 91.0 |
| >8.1 g/dL | 483 | 2.8 | 718 | 3.9 |
| Serum albumin level | | | | |
| <4.1 g/dL | 1,943 | 11.3 | 2,340 | 12.6 |
| 4.1–5.1 g/dL | 15,150 | 88.3 | 16,175 | 87.0 |
| >5.1 g/dL | 67 | 0.4 | 71 | 0.4 |
| Serum gamma gap level | | | | |
| <2.5 g/dL | 3,059 | 17.8 | 1,922 | 10.3 |
| 2.5–3.0 g/dL | 10,689 | 62.3 | 11,455 | 61.6 |
| >3.0 g/dL | 3,412 | 19.9 | 5,209 | 28.0 |
| Proteinuria | | | | |
| ≥± | 1,423 | 8.3 | 747 | 4.0 |
| | Mean ± SD | | | |
| Serum total protein levels (g/dL) | 7.22 ± 0.41 | | 7.30 ± 0.42 | |
| Serum albumin levels (g/dL) | 4.45 ± 0.26 | | 4.43 ± 0.25 | |
| Serum gamma gap levels (g/dL) | 2.77 ± 0.36 | | 2.87 ± 0.35 | |

BMI, body mass index; SD, standard deviation.

increased ORs for low gamma gap levels were observed in current smoking and drinking, green vegetable and coffee intakes, history of colon polyp and fatty liver, and low energy-adjusted protein intake; decreased ORs were observed in men, hypertension, dyslipidemia, and glucose intolerance. Increased ORs for high gamma gap levels were significant in the older groups aged 50–59 and 60–69 years, men, low and high BMI, hypertension, dyslipidemia, glucose intolerance, liver cirrhosis history, high energy-adjusted protein intake and proteinuria. On the other hand, decreased ORs were significant with current smoking and drinking, habitual exercise, high education, intake of fish, green vegetable and coffee, peptic ulcer history. The OR for high gamma gap levels with cirrhosis history (3.36, 2.11–5.35) was apparently higher than the others.

**Table 2. OR and 95% CI for low and high levels of albumin and gamma gap according to various background in the cross-sectional analysis at the baseline.**

| | Albumin | | | | Gamma gap | | | |
|---|---|---|---|---|---|---|---|---|
| | Low | | High | | Low | | High | |
| | OR[a] | 95% CI | OR[a] | 95% CI | OR[a] | 95% CI | OR[a] | 95% CI |
| Age (50–59 vs. 35–49 years) | 1.30 | 1.19-1.43 | 0.83 | 0.54-1.27 | 1.07 | 0.98-1.16 | 1.13 | 1.05-1.22 |
| Age (60–69 vs. 35–49 years) | 1.74 | 1.58-1.92 | 0.32 | 0.19-0.55 | 0.91 | 0.83-1.00 | 1.31 | 1.21-1.41 |
| Sex (men vs. women) | 1.23 | 1.12-1.35 | 1.30 | 0.80-2.09 | 0.66 | 0.61-0.72 | 1.40 | 1.30-1.50 |
| BMI (<18.5 vs. 18-5-24.9 kg/m²) | 0.83 | 0.72-0.95 | 1.78 | 0.90-3.51 | 1.12 | 0.98-1.28 | 1.13 | 1.01-1.27 |
| BMI (≥25.0 vs 18.5–24.9 kg/m²) | 1.23 | 1.13-1.33 | 0.65 | 0.42-1.00 | 0.78 | 0.72-0.85 | 1.23 | 1.16-1.31 |
| Smoking (former vs. never) | 1.05 | 0.95-1.16 | 0.90 | 0.56-1.45 | 1.05 | 0.95-1.15 | 1.00 | 0.92-1.07 |
| Smoking (current vs. never) | 1.37 | 1.24-1.52 | 0.57 | 0.31-1.03 | 2.11 | 1.92-2.31 | 0.56 | 0.51-0.61 |
| Drinking (former vs. never) | 1.12 | 0.90-1.39 | 1.39 | 0.42-4.60 | 1.07 | 0.85-1.33 | 1.07 | 0.90-1.28 |
| Drinking (≥20 g alcohol/day vs. never) | 1.04 | 0.94-1.15 | 2.24 | 1.33-3.76 | 1.26 | 1.14-1.39 | 0.81 | 0.74-0.88 |
| Daily activity (≥18.0 vs. <3.25 METs h/day) | 1.06 | 0.95-1.17 | 0.72 | 0.42-1.23 | 1.00 | 0.90-1.11 | 0.96 | 0.88-1.04 |
| Habitual exercise (≥2.55 vs. <0.06 METs h/day) | 0.72 | 0.66-0.80 | 1.17 | 0.69-1.97 | 1.07 | 0.97-1.17 | 0.89 | 0.82-0.96 |
| Education (≥college vs. <college) | 0.96 | 0.92-1.01 | 1.03 | 0.78-1.35 | 1.03 | 0.98-1.08 | 0.94 | 0.90-0.97 |
| Intake of | | | | | | | | |
| Meat (≥3 vs. <1 times/week) | 0.96 | 0.86-1.06 | 0.89 | 0.51-1.55 | 1.01 | 0.91-1.12 | 0.92 | 0.85-1.00 |
| Fish (≥5 vs. <3 times/week) | 1.04 | 0.93-1.16 | 0.81 | 0.44-1.49 | 0.97 | 0.87-1.09 | 0.90 | 0.82-0.98 |
| Tofu (≥3 vs. <1 times/week) | 1.11 | 1.02-1.22 | 0.80 | 0.50-1.26 | 1.02 | 0.93-1.11 | 1.07 | 0.99-1.14 |
| Green vegetables (≥5 vs. <1 times/week) | 0.93 | 0.83-1.04 | 1.65 | 0.87-3.13 | 1.18 | 1.05-1.32 | 0.86 | 0.78-0.94 |
| Fruit (≥5 vs. <3 times/week) | 0.95 | 0.87-1.05 | 1.36 | 0.87-2.12 | 1.01 | 0.92-1.11 | 0.95 | 0.89-1.02 |
| Coffee (≥2 cups/day vs. <1 time/week) | 0.89 | 0.80-0.98 | 0.65 | 0.40-1.06 | 1.22 | 1.10-1.34 | 0.75 | 0.69-0.81 |
| Hypertension | 0.67 | 0.61-0.72 | 2.59 | 1.65-4.08 | 0.68 | 0.63-0.74 | 1.36 | 1.28-1.45 |
| Dyslipidemia | 0.60 | 0.56-0.65 | 2.45 | 1.65-3.65 | 0.78 | 0.73-0.83 | 1.13 | 1.07-1.19 |
| Glucose intolerance | 1.00 | 0.93-1.07 | 0.57 | 0.38-0.86 | 0.89 | 0.83-0.96 | 1.14 | 1.08-1.20 |
| History of | | | | | | | | |
| Peptic ulcer | 1.14 | 1.05-1.24 | 0.73 | 0.43-1.21 | 1.07 | 0.98-1.15 | 0.92 | 0.85-0.99 |
| Gastritis | 1.02 | 0.92-1.13 | 1.08 | 0.62-1.89 | 1.08 | 0.98-1.20 | 0.94 | 0.86-1.02 |
| Colon polyp | 1.17 | 1.05-1.31 | 1.32 | 0.74-2.33 | 1.15 | 1.03-1.28 | 0.92 | 0.84-1.01 |
| Liver cirrhosis | 3.33 | 2.10-5.28 | 4.35 | 0.55-34.3 | 0.82 | 0.37-1.83 | 3.36 | 2.11-5.35 |
| Fatty liver | 0.68 | 0.60-0.77 | 1.79 | 1.09-2.94 | 1.13 | 1.01-1.26 | 1.07 | 0.98-1.17 |
| Chronic bronchitis | 0.88 | 0.70-1.11 | 0.37 | 0.05-2.72 | 1.18 | 0.96-1.46 | 0.94 | 0.79-1.14 |
| Asthma | 1.03 | 0.90-1.18 | 0.82 | 0.37-1.79 | 1.01 | 0.88-1.15 | 1.01 | 0.91-1.13 |
| Cancer | 1.02 | 0.87-1.19 | 1.75 | 0.86-3.53 | 0.98 | 0.84-1.15 | 1.04 | 0.92-1.17 |
| Energy-adjusted protein intake | | | | | | | | |
| <49.8 vs.49.8–55.4 g/day | 1.00 | 0.92-1.10 | 1.02 | 0.64-1.62 | 1.09 | 1.00-1.18 | 1.02 | 0.95-1.09 |
| >55.4 vs. 49.8–55.4 g/day | 0.91 | 0.82-1.01 | 1.27 | 0.73-2.21 | 0.98 | 0.89-1.09 | 1.10 | 1.01-1.19 |
| Proteinuria (≥ ± vs. -) | 1.61 | 1.42-1.82 | 1.12 | 0.53-2.38 | 1.08 | 0.93-1.24 | 1.38 | 1.24-1.53 |

OR, odds ratio; CI, confidence interval; BMI, body mass index; METs, metabolic equivalents.

[a]ORs and 95% CIs were obtained by multivariate logistic regression analysis using age, sex, BMI, smoking, drinking, daily activity, habitual exercise, education, intake of green vegetables, fruit and coffee, hypertension, dyslipidemia, glucose intolerance, history of peptic ulcer, gastritis, colon polyp, liver cirrhosis, fatty liver, chronic bronchitis, asthma and cancer, energy-adjusted protein intake, and proteinuria as covariates.

The elevated HRs for all-cause death with high total protein, low albumin, and high gamma gap levels were significant in models 1, 2, and 3 (Table 3). The HRs in model 3 were 1.33 (1.05–1.68) with high total protein levels, 1.69 (1.50–1.91) with low albumin levels, and 1.40 (1.25–1.57) with high gamma gap levels.

HRs for deaths from all-causes and specific causes, such as cancer, CVD, RSD without pneumonia, pneumonia, and other-causes, with the combination of albumin and gamma gap levels were estimated (Table 4). Increased HRs for deaths from all-causes, cancer, CVD, RSD without pneumonia, and other-causes were observed in low albumin group with medium and high gamma levels, except other-causes death with medium gamma gap levels (Table 4). The increased HRs with high gamma gap levels were higher in low albumin group than in medium albumin group, particularly for RSD without pneumonia. On the contrary, extremely high HR (12.4, 3.98–38.5) was revealed for pneumonia death with low albumin and low gamma gap group. Increased HRs for death from all-causes, CVD and RSD without pneumonia were also observed in medium albumin group with high gamma gap levels. High HR for other-causes death (4.49, 1.09–18.5) was revealed in high albumin group with high gamma gap levels, too. *P*-values for interaction were statistically significant between low albumin and high gamma gap groups for death from all-causes (*p = 0.003*) and other-causes (*p = 0.010*) and between low albumin and low gamma gap groups for pneumonia death (*p = 0.004*).

The subgroup analysis among those (n = 20,851) without having history of diseases similarly revealed increased HRs for death from all-causes, CVD, RSD without pneumonia, and other-causes with low albumin levels and with high gamma gap levels (Table 5). Increased HR for cancer death was significant in low albumin group, and close to significant in high gamma gap group.

The dose relationship for the HRs with albumin and gamma gap levels were observed using RCSs (Fig 2). The increasing trends on the HRs for deaths from all-causes, cancer, CVD, RSD without pneumonia, pneumonia and other-causes

**Table 3. Hazard ratios and their 95% CIs for all-causes of death according to serum total protein, albumin and gamma gap levels.**

| | All-causes of death | | | | | | | | |
| --- | --- | --- | --- | --- | --- | --- | --- | --- |
| | | | Model 1 | | Model 2 | | Model 3 | |
| | PY | E | HR[a] | 95% CI | HR[b] | 95% CI | HR[c] | 95% CI |
| Total protein | | | | | | | | |
| <6.6 g/dL | 27396 | 128 | 1.29 | 1.07-1.55 | 1.19 | 0.99-1.44 | 1.12 | 0.94-1.36 |
| 6.6–8.1 g/dL | 380395 | 1326 | 1.00 | – | 1.00 | – | 1.00 | – |
| >8.1 g/dL | 14346 | 75 | 1.39 | 1.10-1.76 | 1.37 | 1.08-1.73 | 1.33 | 1.05-1.68 |
| Albumin | | | | | | | | |
| <4.1 g/dL | 51597 | 332 | 1.79 | 1.58-2.02 | 1.74 | 1.54-1.97 | 1.69 | 1.50-1.91 |
| 4.1–5.1 g/dL | 369001 | 1192 | 1.00 | – | 1.00 | – | 1.00 | – |
| >5.1 g/dL | 1538 | 5 | 1.25 | 0.52-3.14 | 1.22 | 0.51-2.95 | 1.13 | 0.47-2.72 |
| Gamma gap | | | | | | | | |
| <2.5 g/dL | 58578 | 194 | 1.09 | 0.93-1.28 | 1.06 | 0.90-1.24 | 1.02 | 0.87-1.19 |
| 2.5–3.0 g/dL | 260705 | 836 | 1.00 | – | 1.00 | – | 1.00 | – |
| >3.0 g/dL | 102853 | 499 | 1.40 | 1.25-1.56 | 1.39 | 1.24-1.56 | 1.40 | 1.25-1.57 |

HR, hazard ratio; CI, confidence interval; E, events; PY, person-years; BMI, body mass index.

[a]Adjusted for age, sex and BMI.

[b]Adjusted for age, sex, BMI, smoking, drinking, daily activity, habitual exercise, education, intake of meat, fish, tofu, green vegetables, fruit and coffee, hypertension, dyslipidemia, glucose intolerance, and history of peptic ulcer, gastritis, colon polyp, liver cirrhosis, fatty liver, chronic bronchitis, asthma and cancer.

[c]Adjusted for age, sex, BMI, smoking, drinking, daily activity, habitual exercise, education, intake of green vegetables, fruit and coffee, hypertension, dyslipidemia, glucose intolerance, history of peptic ulcer, gastritis, colon polyp, liver cirrhosis, fatty liver, chronic bronchitis, asthma and cancer, energy-adjusted protein intake, and proteinuria.

**Table 4. Hazard ratios and their 95% CIs for death from all-causes and specific causes according to the combination of serum albumin and gamma gap levels.**

| | | All-causes | | Cancer | | CVD | | RSD without pneumonia | | Pneumonia | | Other-causes | |
|---|---|---|---|---|---|---|---|---|---|---|---|---|---|
| | PY | E | HRª (95% CI) | E | HRª (95% CI) | E | HRª (95% CI) | E | HRª (95% CI) | E | HRª (95% CI) | E | HRª (95% CI) |
| Low albumin | | | | | | | | | | | | | |
| Low gamma gap | 6028 | 24 | 1.14 (0.76-1.73) | 10 | 0.84 (0.44-1.57) | 5 | 1.67 (0.67-4.15) | 0 | – | 5 | 12.4 (3.98-38.5) | 4 | 0.84 (0.31-2.29) |
| *P* for interaction | | | *0.214* | | *0.073* | | *0.802* | | – | | *0.004* | | *0.676* |
| Medium gamma gap | 28502 | 148 | 1.44 (1.20-1.73) | 76 | 1.41 (1.10-1.81) | 27 | 1.70 (1.11-2.61) | 11 | 2.09 (1.03-4.22) | 2 | 0.73 (0.16-3.34) | 32 | 1.29 (0.88-1.89) |
| High gamma gap | 17068 | 160 | 2.63 (2.21-3.14) | 66 | 2.15 (1.64-2.81) | 24 | 2.45 (1.56-3.85) | 22 | 7.31 (4.15-12.9) | 2 | 1.51 (0.33-6.92) | 46 | 2.99 (2.13-4.19) |
| *P* for interaction | | | *0.003* | | *0.098* | | *0.911* | | *0.491* | | *0.385* | | *0.010* |
| Medium albumin | | | | | | | | | | | | | |
| Low gamma gap | 52349 | 170 | 1.08 (0.91-1.28) | 100 | 1.15 (0.91-1.44) | 27 | 1.14 (0.75-1.76) | 8 | 1.20 (0.54-2.66) | 4 | 1.18 (0.38-3.69) | 31 | 0.83 (0.57-1.23) |
| Medium gamma gap | 231388 | 686 | Reference | 365 | Reference | 112 | Reference | 30 | Reference | 15 | Reference | 164 | Reference |
| High gamma gap | 85264 | 336 | 1.22 (1.07-1.39) | 150 | 1.09 (0.90-1.32) | 68 | 1.40 (1.03-1.91) | 33 | 2.55 (1.54-4.22) | 5 | 0.73 (0.26-2.07) | 80 | 1.16 (0.88-1.52) |
| High albumin | | | | | | | | | | | | | |
| Low gamma gap | 201 | 0 | – | 0 | – | 0 | – | 0 | – | 0 | – | 0 | – |
| *P* for interaction | | | – | | – | | – | | – | | – | | – |
| Medium gamma gap | 815 | 2 | 0.92 (0.23-3.72) | 1 | 0.79 (0.11-5.65) | 0 | – | 0 | – | 0 | – | 1 | 2.23 (0.31-16.0) |
| High gamma gap | 521 | 3 | 1.75 (0.56-5.48) | 1 | 0.98 (0.14-7.04) | 0 | – | 0 | – | 0 | – | 2 | 4.49 (1.09-18.5) |
| *P* for interaction | | | *0.567* | | *0.903* | | – | | – | | – | | *0.518* |

CVD, cardiovascular diseases; RSD, respiratory system diseases; E, events; PY, person-years; HR, hazard ratio; CI, confidence interval; BMI, body mass index.

Levels of albumin: low, < 4.1 g/dL; medium, 4.1–5.1 g/dL; high, > 5.1 g/dL. Levels of gamma gap: low, < 2.5 g/dL; medium, 2.5–3.0 g/dL; high, > 3.0 g/day.

a) Adjusted for age, sex, BMI, daily activity, habitual exercise, education, intake of green vegetables, fruit and coffee, hypertension, dyslipidemia, glucose intolerance, history of peptic ulcer, gastritis, colon polyp, liver cirrhosis, fatty liver, chronic bronchitis, asthma and cancer, energy-adjusted protein intake, and proteinuria.

with decreasing albumin levels were dose-dependent, especially at lower levels than lower limit of albumin standard value (4.1 mg/dL). The relationship between the HRs for deaths from all-causes, cancer, CVD, RSD without pneumonia and other-causes, and gamma gap levels was threshold-type, and their HRs increased above higher levels than higher limit of gamma gap standard value (3.0 mg/dL), except pneumonia. The relationship between HR for pneumonia death and gamma gap levels showed an opposite trend; however, the HR was not statistically significant. The relationship between HR for other-causes death and gamma gap levels was dose-dependent.

## Discussion

This cohort study investigated the associations between serum albumin and gamma gap levels and their combined effect, and mortality risk in a Japanese population. Elevated HRs for deaths from all-causes, cancer, CVD, RSD without pneumonia, pneumonia, and other causes were independently observed in the low/high albumin and gamma gap level groups after

**Table 5. Hazard ratios and their 95% CIs for death from all-causes and specific causes according to serum albumin and gamma gap levels among those without history of related-diseases[a].**

| | | All-causes | | | Cancer | | | CVD | | | RSD without pneumonia | | | Pneumonia | | | Other-causes | | |
|---|---|---|---|---|---|---|---|---|---|---|---|---|---|---|---|---|---|---|---|
| | PY | E | HR[b] | 95% CI | E | HR[b] | 95% CI | E | HR[b] | 95% CI | E | HR[b] | 95% CI | E | HR[b] | 95% CI | E | HR[b] | 95% CI |
| Albumin | | | | | | | | | | | | | | | | | | | |
| <4.1 g/dL | 29805 | 157 | 1.67 | 1.40-2.00 | 66 | 1.41 | 1.08-1.85 | 29 | 1.83 | 1.20-2.78 | 19 | 3.32 | 1.89-5.86 | 4 | 2.97 | 0.88-10.0 | 39 | 1.61 | 1.12-2.30 |
| 4.1–5.1 g/dL | 219310 | 617 | 1.00 | – | 305 | 1.00 | – | 108 | 1.00 | – | 37 | 1.00 | – | 10 | 1.00 | – | 157 | 1.00 | – |
| >5.1 g/dL | 897 | 2 | 0.97 | 0.24-3.90 | 1 | 0.96 | 0.13-6.89 | 0 | – | – | 0 | – | – | 0 | – | – | 1 | 1.77 | 0.25-12.7 |
| Gamma gap | | | | | | | | | | | | | | | | | | | |
| <2.5 g/dL | 31094 | 86 | 1.07 | 0.85-1.36 | 50 | 1.17 | 0.85-1.60 | 14 | 1.17 | 0.65-2.10 | 3 | 0.78 | 0.23-2.66 | 3 | 2.07 | 0.49-8.85 | 16 | 0.79 | 0.46-1.34 |
| 2.5–3.0 g/dL | 154808 | 416 | 1.00 | – | 210 | 1.00 | – | 71 | 1.00 | – | 21 | 1.00 | – | 7 | 1.00 | – | 107 | 1.00 | – |
| >3.0 g/dL | 64110 | 274 | 1.46 | 1.25-1.70 | 112 | 1.25 | 0.99-1.58 | 52 | 1.51 | 1.04-2.18 | 32 | 3.21 | 1.83-5.64 | 4 | 1.36 | 0.38-4.88 | 74 | 1.47 | 1.09-2.00 |

CVD, cardiovascular diseases; RSD, respiratory system diseases; E, events; PY, person-years; HR, hazard ratio; CI, confidence interval; BMI, body mass index.

[a])Number of participants: 20851.

[b])Adjusted for age, sex, BMI, daily activity, habitual exercise, education, intake of green vegetables, fruit and coffee, hypertension, dyslipidemia, glucose intolerance, energy-adjusted protein intake, and proteinuria.

adjusting for confounding factors. Especially, low albumin and low gamma gap levels were strongly associated for pneumonia death. The combined effect was also apparent in low albumin group with low/high gamma gap levels. The dose relationship for each association was dose-dependent in albumin and threshold-type in gamma gap, except other-causes.

## Serum albumin

Although there may be few cases in the general population that have a background of large changes in albumin levels like as clinical patients, the present baseline cross-sectional observation showed that low serum albumin levels were associated with various backgrounds. The details are unclear, but various factors and the pathophysiology of albumin synthesis, catabolism, and loss have been suggested to be associated with low serum albumin levels even in the general population. The present cohort study showed that low serum albumin levels with medium gamma gap levels were independently associated with elevated HRs for deaths with a clear dose-dependent relationship, even after adjusting for related-factors. This suggests that low serum albumin levels are associated with an increased risk of mortality not only in clinical patients [25,26] but also in the general population. As the increase in HRs for mortality was observed after adjusting for protein intake and proteinuria, the effects of insufficient protein intake and loss from urine are considered small. It cannot be ruled out that hypo-albuminemia itself may relate to an increased risk of death, because one of the roles of albumin is to bind to endogenous substances [4], such as maintaining a normal nitric oxide level [27]. Several mechanisms may be involved in the roles of albumin, and it has not yet been clarified.

In contrast, high baseline serum albumin levels were associated with factors related to metabolic syndrome such as alcohol consumption, hypertension, dyslipidemia, and fatty liver. The inverse association between older age, glucose intolerance, and high serum albumin levels suggests a relationship with diet. There was no association between high serum albumin levels with medium gamma gap levels and the adjusted HRs for death, partly because fewer participants had elevated serum albumin levels.

## Gamma gap

Gamma gap is used as a clinical screening index for latent inflammation, cancer, and autoimmune diseases [7–10]. High serum globulin levels in US life insurance insured individuals were also associated with high risk of all-cause mortality [28].

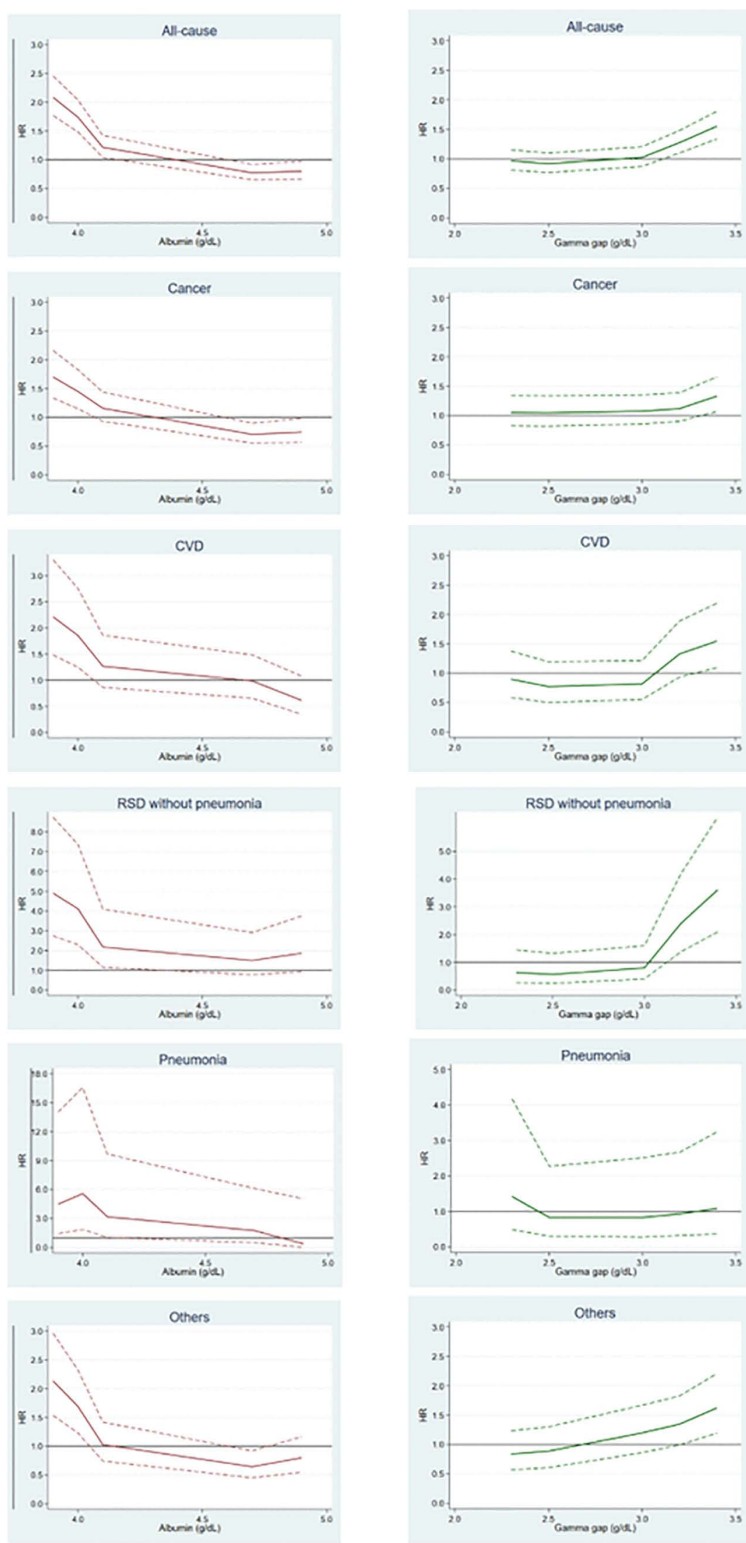

**Fig 2. RCS on albumin and gamma gap levels, and death HR.** RCS showed a spline curve combined cubic polynomials and linear terms. Solid line is adjusted HR, and dashed line is 95% CI. The values of the knots in albumin are 3.9, 4.0, 4.1, 4.7, and 4.9 g/dL, and the reference is 4.4 g/dL. The values of the knots in gamma gap are 2.3, 2.5, 3.0, 3.2 and 3.4 g/dL, and the reference is 2.8 g/dL. The lower values than the lowest knot and the higher values

than the highest knot were respectively truncated in the nearest section. RCS, restricted cubic splines; HR, hazard ratio; CI, confidence interval; CVD, cardiovascular disease; RSD, respiratory system diseases.

Various factors which were associated with gamma gap levels in cross-sectional observation are concordant with those for latent inflammation, cancer, and autoimmune diseases, in which the gamma gap is used as a clinical screening index [7–10].

The present study showed that elevated serum gamma gap levels with medium albumin levels were independently associated with increased HRs for deaths of all-causes, CVD, and RSD without pneumonia, after adjusting for related factors. A previous study with a general population in the United States reported that gamma gap was positively associated with mortality risk [13]. The cut-off value for gamma gap levels in this previous report was 3.1 g/dL, which was the same as the > 3.0 g/dL in the present study. In addition, the HR with high gamma gap levels for death from RSD without pneumonia was higher than the HRs for mortality of all-causes, cancer, CVD, and other-causes, which is consistent with the present results [13]. This study suggests that high serum gamma gap levels are an indicator of a high risk for mortality due to various diseases in a Japanese population, which may be involved in chronic inflammation across the lifespan [29].

## Combination of albumin and gamma gap

Although levels of albumin and gamma gaps had a weak negative relationship, this study showed an apparent combined effect on the HRs for death between albumin and gamma gap levels. The HRs for any deaths, except pneumonia, with low albumin and high gamma gap levels was higher than those with low albumin and medium gamma gap levels, and the HR was the highest for RSD without pneumonia. Their interaction was significant for deaths from all-causes and other-causes. The mechanism is unknown, but this finding may suggest high risk population for death with low albumin levels include further higher risk people with high gamma gap levels. On the other hand, low albumin and low gamma gap levels were strongly associated for pneumonia death, and significant interaction was also observed between low albumin and low gamma gap levels for pneumonia death. Low gamma gap levels mean week immunological background, especially for infectious diseases, and this study suggested low albumin levels further elevated its risk.

## Strengths and limitations

The strength of this study is that it simultaneously clarified the association between serum albumin and gamma gap levels, and mortality risk in an Asian population other than clinical patient population, which combined effect has not been previously reported. Furthermore, protein intake and proteinuria, which may affect serum protein levels, were added as confounding variables.

However, this study has some limitations. First, it should be noted that low albumin and high gamma gap levels may be intermediate markers caused by various causes and conditions. In addition, the background factors related to high and low levels of albumin and gamma gap at baseline are diverse and require individual consideration. Second, we investigated the association of major causes of death, such as cancer, CVD, RSD, and other causes; however, a more detailed sub-analysis by disease could not be performed due to a lack of statistical power because of the limited number of deaths. Third, adjustments were made for various factors, including protein intake and proteinuria; however, the effects of residual confounding factors remain unknown. Forth, the possibility of causal reversal remains. To reduce this effect, dead cases within two years from the baseline were excluded from the analysis, and the history of diseases related to low/high albumin and gamma gap levels was adjusted to estimate the HRs for death. Furthermore, sub-group analysis among those without history of related-diseases was done, and the similar HRs were concordantly observed. Fifth, as gamma gap includes various kinds of proteins from a comprehensive metabolic panel, its sensitivity and specificity compared to albumin alone may be low. Sixth, the present study population may not be representative of Japanese general population, although they were recruited from residents in 10 study regions.

## Conclusion

This cohort study investigated the associations between serum albumin and gamma gap levels, and their combined effect, and risk of deaths from all-causes, cancer, CVD, RSD without pneumonia, pneumonia, and other-causes in a Japanese population. Background factors associated with high and low levels of serum albumin and gamma gap are diverse, and their pathophysiology and causality need to be examined in more detail according to disease and individual. In conclusion, this study suggests that albumin and gamma gap levels are independent indicators of an increased risk of mortality in a Japanese population. Combined effect was apparent for mortality from all-causes, pneumonia, and other-causes. The population with low albumin or high gamma gap even without specific diseases may be required careful follow-up in health checkup. Especially, particular attention may be given to people with low albumin and high gamma gap for RSD without pneumonia, and low albumin and low gamma gap for pneumonia.

## Supporting information

**S1 File. Full membership list of the author group.**
(DOCX)

## Acknowledgments

The authors thank Drs. Nobuyuki Hamajima and Hideo Tanaka for initiating and organizing the J-MICC study as former principal investigators. The current principal investigator is Keitaro Matsuo, MD, PhD, Division of Cancer Epidemiology and Prevention, Aichi Cancer Center, Nagoya, Japan (kmatsuo@aichi-cc.jp). All authors were within J-MICC study group, except Drs. Shuhei Niiyama and Yasuyuki Kakihana, Department of Emergency and Intensive Care Medicine, Kagoshima University Graduate School of Medical and Dental Sciences. We are grateful to the research participants and members of the J-MICC Study Group who are listed in the supplementary file.

## Author contributions

**Conceptualization:** Kenichi Shibuya, Toshiro Takezaki.

**Data curation:** Mako Nagayoshi, Takashi Tamura, Yudai Tamada, Rieko Okada.

**Formal analysis:** Kenichi Shibuya, Toshiro Takezaki.

**Investigation:** Rie Ibusuki, Daisaku Nishimoto, Toshiro Takezaki, Megumi Hara, Yuichiro Nishida, Sadao Suzuki, Takeshi Nishiyama, Mako Nagayoshi, Takashi Tamura, Yudai Tamada, Rieko Okada, Teruhide Koyama, Satomi Tomida, Kiyonori Kuriki, Jun Otonari, Hiroaki Ikezaki, Asahi Hishida, Masashi Ishizu, Sakurako Katsuura-Kamano, Kenji Wakai, Keitaro Matsuo.

**Project administration:** Kenji Wakai.

**Supervision:** Keitaro Matsuo.

**Visualization:** Kenichi Shibuya, Toshiro Takezaki.

**Writing – original draft:** Kenichi Shibuya.

**Writing – review & editing:** Shiroh Tanoue, Chihaya Koriyama, Shuhei Niiyama, Yasuyuki Kakihana, Toshiro Takezaki.

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
