## [Decision Letter · Decision Letter 0]

25 Aug 2025

Dear Dr. Takezaki,

Thank you for submitting your manuscript to PLOS ONE. After careful consideration, we feel that it has merit but does not fully meet PLOS ONE’s publication criteria as it currently stands. Therefore, we invite you to submit a revised version of the manuscript that addresses the points raised during the review process.

We look forward to receiving your revised manuscript.

Kind regards,

Nayan Chandra Mohanto, Ph.D.

Academic Editor

PLOS ONE

Journal Requirements:

“This study was supported by Grants-in-Aid for Scientific Research for Priority Areas of Cancer [grant number 17015018], Innovative Areas [grant number 221S0001], and the Japan Society for the Promotion of Science (JSPS) KAKENHI [grant numbers. 16H06277 and 22H04923; CoBiA] from the Japanese Ministry of Education, Culture, Sports, Science, and Technology.”

3. Please note that funding information should not appear in the Acknowledgments section or other areas of your manuscript. We will only publish funding information present in the Funding Statement section of the online submission form. Please remove any funding-related text from the manuscript. 

5. In the online submission form, you indicated that your data is available only on request from a third party. Please note that your Data Availability Statement is currently missing the contact details for the third party, such as an email address or a link to where data requests can be made. Please update your statement with the missing information.

6. One of the noted authors is a group or consortium: J-MICC Study group

In addition to naming the author group, please list the individual authors and affiliations within this group in the acknowledgments section of your manuscript. Please also indicate clearly a lead author for this group along with a contact email address.

**Additional Editor Comments:**

Dear author (s),

Thank you for submitting your manuscript in PLOS one. We are regret informing you that the present version of your manuscript is not suitable for publications. Please go through the reviewers comments and try to address all the suggestions. Please ensure that language in the manuscript is grammatically correct. Explain the methodology more clearly. Improve the scientific flaw of writing highlighting the novelty of the study. Discussion part should be more specific based on your findings. Please resubmit your revised manuscript once it will be ready.

With thanks

Academic editor

Reviewers' comments:

Reviewer's Responses to Questions

**Comments to the Author**

1. Is the manuscript technically sound, and do the data support the conclusions?

Reviewer #1: Yes

Reviewer #2: Yes

2. Has the statistical analysis been performed appropriately and rigorously?

Reviewer #1: I Don't Know

Reviewer #2: Yes

3. Have the authors made all data underlying the findings in their manuscript fully available?

Reviewer #1: No

Reviewer #2: No

4. Is the manuscript presented in an intelligible fashion and written in standard English?

Reviewer #1: Yes

Reviewer #2: Yes

Reviewer #1: This cohort study explored the associations between serum albumin and gamma gap levels as well as their combined effect on mortality risk in a Japanese population. There are so many variables in this study to compare among low to high albumin levels, protein intake, total protein level, and gamma gap levels with all-causes and specific causes of death. Initially readers may feel a bit uncomfortable to understand the whole process and analysis. However the authors involved in this study have been written the manuscript in a very clear and understandable manner. They also very clearly identified their limitations. I truly appropriate their efforts and honesty. Anyway there are some minor correction to improve this article. The comments are listed below:

1.My first impression about the co-authorship of this parer. I found, there are 26 co-authors in this paper. Although this study includes huge number of participants around 36,000, I can understand the contributions of each authors. However, I would suggest you to reduce the number of co-authors in this paper. Previously published one cohort study in PLOS One journal that analyzed more than 80,000 samples were just 10 co-authors.

By considering this study, I would suggest to keep maximum 15-18 co-authors.

2.In Table 1, Table 3, and Table 5, please correct the unit of serum high gamma gap levels.

3.In Table 3, please correct the spelling mistake Medel 1, Medel 2 and Medel 3 on the top of the table.

Reviewer #2: The following list of issues needs to be addressed:

1. Throughout the manuscript some statements lack of reference. You should add more updated reference.

2. Discuss further details about the function of albumin how it maintains the oncotic pressure as well as how it contributes to transport, solubilization and stabilization.

3. The manuscript would be more understandable after the inclusion of a flow diagram of study design that outlines participant recruitment, inclusion and exclusion criteria, and the ultimate sample size.

4. The measurement technique and kits used for the quantification of total serum protein and albumin are not described. Clarify the information in the materials and methods section.

5. What criteria did you use to separate the age groups of 35–49, 50–59, and 60–69 years? In the statistical part, explain it.

6. Mean or median of serum total protein, albumin, and gamma gap level for male and female is necessary for a better understanding.

7. What type of information you are providing or recommending to the general population through this study.

8. Please briefly discuss the physiological and clinical significance of low albumin and high levels of gamma gap level in respect to the diseases.

9. There have repetitions of some information and typos error. You should revise the manuscript properly.

10. Show the relationship among serum total protein, albumin, and gamma gap levels, if there have.

**Do you want your identity to be public for this peer review?** For information about this choice, including consent withdrawal, please see our Privacy Policy

Reviewer #1: No

Reviewer #2: No

---

## [Author Response · Author response to Decision Letter 1]

2 Oct 2025

Dear Reviewer 1:

On behalf of my co-authors, we would like to express our great appreciation to the reviewers. We have carefully taken the reviewers’ comments into consideration in the preparation of our revision. All changes to the reviewer 1 and reviewer 2 are combined and are highlighted in yellow maker in the manuscript. We have included point-to-point response to the reviewer’ comments as follows.

1. My first impression about the co-authorship of this paper. I found, there are 26 co-authors in this paper. Although this study includes huge number of participants around 36,000, I can understand the contributions of each authors. However, I would suggest you to reduce the number of co-authors in this paper. Previously published one cohort study in PLOS One journal that analyzed more than 80,000 samples were just 10 co-authors. By considering this study, I would suggest to keep maximum 15-18 co-authors.

We can understand reviewer’s suggestion. However, our J-MICC study group has our authorship rule including number of authors according to each contribution by institute and university. Our group has previously published manuscripts to the PLoS One according to our rule: PLoS One. 2023;18(2):e0279169. doi: 10.1371/journal.pone.0279169 with 32 authors; and PLoS One. 2022;17(1):e0262252. doi: 10.1371/journal.pone.0262252 with 29 authors.　We would appreciate it if you could let us keep 26 authors.

2. In Table 1, Table 3, and Table 5, please correct the unit of serum high gamma gap levels.

We correct the unit of serum high gamma gap levels in Table 1, Table 3, and Table 5, and Fig 2, too.

3. In Table 3, please correct the spelling mistake Medel 1, Medel 2 and Medel 3 on the top of the table.

We correct the spelling mistake to Model 1, Model 2 and Model 3 in Table 3.

Dear Reviewer 2:

On behalf of my co-authors, we would like to express our great appreciation to the reviewers. We have carefully taken the reviewers’ comments into consideration in the preparation of our revision. All changes to the reviewer 1 and reviewer 2 are combined and are highlighted in yellow maker in the manuscript. We have included point-to-point response to the reviewer’ comments as follows.

1. Throughout the manuscript some statements lack of reference. You should add more updated reference.

We added and updated references. We also deleted the sentences without reference in Discussion.

[update]

1. Henry's Clinical Diagnosis and Management by Laboratory Methods. 24 ed. Philadelphia, PA: Elsevier; 2021.

2. Serum Albumin: Structure, Functions and Health Impact (Protein Biochemistry, Synthesis, Structure and Cellular Functions). New York: Nova Science Pub Inc; 2012.

[add]

4. Moramarco S, Morciano L, Morucci L, Messinese M, Gualtieri P, Carestia M. Epidemiology of Hypoalbuminemia in Hospitalized Patients: A Clinical Matter or an Emerging Public Health Problem? Nutrients. 2020 Nov 27;12(12):3656. doi: 10.3390/nu12123656. PubMed: PMC7760225.

5. Oster HS, Dolev Y, Kehat O, Weis-Meilik A, Mittelman M. Serum Hypoalbuminemia Is a Long-Term Prognostic Marker in Medical Hospitalized Patients, Irrespective of the Underlying Disease. J Clin Med. 2022;11(5):1207. doi: 10.3390/jcm11051207. PubMed: PMC8911288.

2. Discuss further details about the function of albumin how it maintains the oncotic pressure as well as how it contributes to transport, solubilization and stabilization.

We added further details about the function of albumin in Introduction.

Introduction, line 59-77; “Albumin is a negatively charged protein with a high concentration and relatively large molecular weight, accounting for approximately 75–80% of plasma colloid osmotic pressure [2]. The negative charge of the albumin molecule also attracts cations and water, contributing to water retention within capillaries and playing a crucial role in stabilizing blood volume [2]. Furthermore, albumin molecules possess various ligand-binding sites, enabling them to bind and transport substances such as fatty acids, hormones, bilirubin, and calcium [2]. Approximately 40% of the calcium in whole blood exists in a bound state with albumin, which buffers and stabilizes blood calcium levels [1]. Highly hydrophobic molecules also become soluble in blood by binding to albumin, allowing them to be transported to target tissues [2]. Furthermore, albumin binds to drugs and toxins, suppressing the concentration of free active forms and mitigating rapid onset of action or toxicity [2]. Physiologically, low albumin levels impair the maintenance of adequate plasma colloidal osmolarity. This makes it easier for fluid to leak from blood vessels into tissues, potentially causing edema, pleural effusion, and ascites. Furthermore, as albumin transports various substances, its reduction serves as an indicator of conditions such as malnutrition, liver dysfunction, and chronic inflammation. Clinically, hypoalbuminemia is a poor prognostic indicator, with numerous studies showing a strong association with increased mortality in hospitalized patients and those with chronic diseases [4, 5].”

3. The manuscript would be more understandable after the inclusion of a flow diagram of study design that outlines participant recruitment, inclusion and exclusion criteria, and the ultimate sample size.

We added a flow diagram in Fig 1.

Figure Legend, page 45; Fig 1. Flow diagram for selecting eligible participants from 13 cohorts.

4. The measurement technique and kits used for the quantification of total serum protein and albumin are not described. Clarify the information in the materials and methods section.

We added the measurement technique and kits used for the quantification of total serum protein and albumin in Methods/Clinical characteristics.

Methods/Clinical characteristics, line 162-165; “Examination of serum total protein levels was carried out using the Biuret method. Serum albumin levels were examined using bromocresol green (BCG) test or modified bromocresol purple (BCP) test.”

5. What criteria did you use to separate the age groups of 35-49, 50-59, and 60-69 years? In the statistical part, explain it.

We added the criteria of separated age group in Methods/Statistical analyses.

Methods/Statistical analyses, line 183-184; “Age was divided into three groups (35–49, 50–59, and 60–69 years) according to similar number of participants.”

6. Mean or median of serum total protein, albumin, and gamma gap level for male and female is necessary for a better understanding.

We added the mean and standard deviation of serum total protein, albumin, and gamma gap level criteria of separated age group in Results and Table 1.

Results, line 250-252; “The mean levels of serum total protein, albumin and gamma gap in men and women were 7.22 and 7.30; 4.45 and 4.43; and 2.77 and 2.87, respectively.”

7. What type of information you are providing or recommending to the general population through this study.

We added the information providing or recommending to the general population through this study in Conclusion. We also added the description on its related results in Abstract and Discussion.

Conclusion, line 417-421; “The population with low albumin or high gamma gap even without specific diseases may be required careful follow-up in health checkup. Especially, particular attention may be given to people with low albumin and high gamma gap for RSD without pneumonia, and low albumin and low gamma gap for pneumonia.”

Abstract, line 18-22; “Increased HRs of low albumin and high gamma gap levels were respectively observed for deaths from all-causes, cancer, cardiovascular diseases, respiratory system diseases without pneumonia, and other-causes; and the HR was the highest on respiratory system diseases without pneumonia (HR = 7.31, 4.15-12.9). Low albumin and low gamma gap levels were strongly associated for pneumonia death (HR = 12.4, 3.98-38.5).”

Discussion, line 371-373; “The HRs for any deaths, except pneumonia, with low albumin and high gamma gap levels was higher than those with low albumin and medium gamma gap levels, and the HR was the highest for RSD without pneumonia.”

Discussion, line 376-379; “On the other hand, low albumin and low gamma gap levels were strongly associated for pneumonia death, and significant interaction was also observed between low albumin and low gamma gap levels for pneumonia death.”

8. Please briefly discuss the physiological and clinical significance of low albumin and high levels of gamma gap level in respect to the diseases.

We added the description on the physiological and clinical significance of low albumin and high levels of gamma gap level in respect to the diseases, although the present results did not suggest the risk of specific diseases, except pneumonia.

Introduction, line 51-58; “Therefore, in addition to insufficient nutritional intake, serum albumin levels decline due to impaired absorption of amino acids because of a breakdown in the gastrointestinal mucosal epithelium, decreased synthesis capacity in the liver from cirrhosis, protein leakage from the gastrointestinal tract and kidneys due to protein-leaking gastroenteropathy and nephrotic syndrome, and hyper-catabolism from severe infections and chronic inflammatory diseases. Among them, production disorder has a great influence on a serum albumin decrease, which depends on the ability of the liver to synthesize [3].

Introduction, line 91-97; “Physiologically, a high gamma gap suggests increased immunoglobulin levels. This indicates the possibility of ongoing chronic inflammation within the body or abnormal plasma cell proliferation. Clinically, it can serve as a differential diagnostic indicator for conditions such as multiple myeloma, macroglobulinemia, and primary amyloidosis. It is also used as a screening indicator for plasma cell disorders, and when a marked elevated gamma gap is observed, further investigation such as protein electrophoresis is recommended.”

Discussion, line 352-354; “Various factors which were associated with gamma gap levels in cross-sectional observation are concordant with those for latent inflammation, cancer, and autoimmune diseases, in which the gamma gap is used as a clinical screening index [7-10].”

Discussion, line 363-366; “This study suggests that high serum gamma gap levels are an indicator of a high risk for mortality due to various diseases in a Japanese population, which may be involved in chronic inflammation across the lifespan [29].”

9. There have repetitions of some information and typos error. You should revise the manuscript properly.

We deleted the parts of information and corrected typos error.

10. Show the relationship among serum total protein, albumin, and gamma gap levels, if there have.

We added the analysis of the relationship among serum total protein, albumin, and gamma gap levels and added it in Methods//Statistical analyses, Results and Discussion.

Methods//Statistical analyses, line 235-236; “The relationship among serum total protein, albumin, and gamma gap levels were also tested using correlation coefficients.”

Results, line 252-256; “The correlation coefficients between serum total protein levels, and albumin and gamma gap levels were 0.523 (p<0.001) and 0.777 (p<0.001) in men, and 0.536 (p<0.001) and 0.794 (p<0.001) in women, respectively (data not shown in Table). The correlation coefficients between albumin and gamma gap levels were -0.130 (p<0.001) in men and -0.088 (p<0.001) in women.”

Discussion, line 369-371; “Although levels of albumin and gamma gaps had a weak negative relationship, this study showed an apparent combined effect on the HRs for death between albumin and gamma gap levels.”

---

## [Decision Letter · Decision Letter 1]

2 Nov 2025

Serum albumin and gamma gap levels, and combined effect for risk of mortality in a Japanese population from the J-MICC study

PONE-D-24-16646R1

Dear Dr. Takezaki,

We’re pleased to inform you that your manuscript has been judged scientifically suitable for publication and will be formally accepted for publication once it meets all outstanding technical requirements.

Kind regards,

Nayan Chandra Mohanto, Ph.D.

Academic Editor

PLOS ONE

Additional Editor Comments (optional):

Dear author (s),

Please address the comments from the reviewer 2 and resubmit the revised manuscript.

Reviewers' comments:

Reviewer's Responses to Questions

**Comments to the Author**

Reviewer #1: All comments have been addressed

Reviewer #2: All comments have been addressed

2. Is the manuscript technically sound, and do the data support the conclusions?

Reviewer #1: Yes

Reviewer #2: Yes

3. Has the statistical analysis been performed appropriately and rigorously?

Reviewer #1: I Don't Know

Reviewer #2: Yes

4. Have the authors made all data underlying the findings in their manuscript fully available?

Reviewer #1: No

Reviewer #2: No

5. Is the manuscript presented in an intelligible fashion and written in standard English?

Reviewer #1: Yes

Reviewer #2: Yes

Reviewer #1: No more additional comments for the authors from my side.

I would like to appreciate all the authors to have their efforts and patients go through our whole reviewing process.

Overall the manuscript is now well written and technically sounds with a better version.

Reviewer #2: 1. It would be better to make the Fig 1 as a supplementary figure.

2. Could you please clarify whether the differences in total protein, albumin, and gamma gap levels between men and women were significant?

3. “Examination of serum total protein levels was carried out using the Biuret method. Serum albumin levels were examined using bromocresol green (BCG) test or modified bromocresol purple (BCP) test.” these two sentences can be combined and written in a following way-

“Examination of serum total protein and albumin levels were carried out using the Biuret method and bromocresol green (BCG) test or modified bromocresol purple (BCP) test, respectively.”

**Do you want your identity to be public for this peer review?** For information about this choice, including consent withdrawal, please see our Privacy Policy

Reviewer #1: **Yes: ** Dr. Momotaj Jahan

Reviewer #2: No

---

## [Editor Report · Acceptance letter]

PONE-D-24-16646R1

PLOS ONE

Dear Dr. Takezaki,

I'm pleased to inform you that your manuscript has been deemed suitable for publication in PLOS ONE. Congratulations! Your manuscript is now being handed over to our production team.

Kind regards,

on behalf of

Dr. Nayan Chandra Mohanto

Academic Editor

PLOS ONE